# Assessment of deep learning image reconstruction (DLIR) on image quality in pediatric cardiac CT datasets type of manuscript: Original research

**Hyun-Hae Cho** [1]*, **So Mi Lee**[2], **Sun Kyoung You** [3]

1 Department of Radiology and Medical Research Institute, College of Medicine, Ewha Womans University Seoul Hospital, Seoul, Republic of Korea, 2 Department of Radiology, School of Medicine, Kyungpook National University, Kyungpook National University Hospital, Daegu, South Korea, 3 Department of Radiology, Chungnam National University Hospital, Daejeon, Republic of Korea

* picoai@yahoo.co.kr

**Data Availability Statement:** All relevant data are within the manuscript and its Supporting information files.

## Abstract

### Bakground

To evaluate the quantitative and qualitative image quality using deep learning image reconstruction (DLIR) of pediatric cardiac computed tomography (CT) compared with conventional image reconstruction methods.

### Methods

Between January 2020 and December 2022, 109 pediatric cardiac CT scans were included in this study. The CT scans were reconstructed using an adaptive statistical iterative reconstruction-V (ASiR-V) with a blending factor of 80% and three levels of DLIR with TrueFidelity (low-, medium-, and high-strength settings). Quantitative image quality was measured using signal-to-noise ratio (SNR) and contrast-to-noise ratio (CNR). The edge rise distance (ERD) and angle between 25% and 75% of the line density profile were drawn to evaluate sharpness. Qualitative image quality was assessed using visual grading analysis scores.

### Results

A gradual improvement in the SNR and CNR was noted among the strength levels of the DLIR in sequence from low to high. Compared to ASiR-V, high-level DLIR showed significantly improved SNR and CNR ($P<0.05$). ERD decreased with increasing angle as the level of DLIR increased.

### Conclusion

High-level DLIR showed improved SNR and CNR compared to ASiR-V, with better sharpness on pediatric cardiac CT scans.

**Funding:** This work was supported by the Ewha Womans University Research Grant of 2021. The funders had no role in study design or data collection. But the financial support for research, including statistical advice, translation advice, and research, is based on the funder.

**Competing interests:** There exist no other competing interests.

**Abbreviations:** ASiR-V, adaptive statistical iterative reconstruction-V; CNR, contrast-to-noise ratio; CT, computed tomography; DLIR, deep learning image reconstruction; ERD, edge rise distance; HU, Hounsfield unit; LV, left ventricle; PACS, picture archiving and communication systems; ROI, region of interest; RV, right ventricle; SNR, signal-to-noise ratio; VGA, visual grading analysis.

# Background

Congenital heart diseases (CHD) are rare diseases and their prognosis is difficult to predict, but an understanding of the exact anatomical structure and hemodynamic function is necessary to determine subsequent treatment and provide proper medical support [1–3]. For evaluation, echocardiography is yet to be used as primary tool in neonates with CHD [1, 3], but echocardiography has limitations in the great vessels and coronary arteries [4]. In terms of the postoperative status, acoustic shadowing can occur due to surgical materials, and echocardiography may not be able to obtain an image suitable for exact evaluation [4].

Recently, the usage of computed tomography (CT) is widely increased in evaluating CHD. It was proved CT showed similar accuracy compared with echocardiography [2–6].And it shows advantages in limited situations to evaluating the complex heart anatomy, postoperative changes and ventricle functions compared with using only echocardiography [1, 2, 4–8]. Recently, the incidence of adulthood-diagnosed congenital heart disease has increased [1, 2, 4, 6], which is hard to evaluated by using echocardiography. For these reasons, CT is now used as important tool for evaluating CHD.

Actually, it is hard to obtain qualified pediatric CT due to motion artifacts caused by the limitations of respiratory hold and higher heart rate. For more accurate delineation of the small complex anatomy is required for pediatric cardiac CT scans, improving the image quality of pediatric cardiac CT is important for reducing unnecessary radiation exposure for relatively radiation sensitive pediatric age group [9–11].

For increasing image quality of pediatric cardiac CT scan, many post processing reconstruction tools were adapted [8, 12–15]. Recently, the latest developed image reconstruction tool is released by using adaptation of deep-learning techniques highlighted in the medical imaging field [13, 16–20]. It has been proven to be effective for improving the image quality of many body parts [16–20]. Therefore, the purpose of this study was to evaluate the image quality using deep learning image reconstruction (DLIR) of pediatric cardiac CT compared to conventional IR methods.

# Methods

Approval from the institutional review board was obtained (IRB number 2022-02-026-010), and the requirement for patient/parent informed consent was waived.

## Imaging

Existing cardiac CT scans were examined with same CT machine (512-slice CT scan; Revolution; GE Healthcare, Milwaukee, USA). We used retrospectively electrocardiography-gated spiral scan for achieve both systolic and diastolic phase images for volume measurement and calculation of EF. Lowest possible tube voltage was decided between 80Kv or 100Kv according to the patient's age and weight. Contrast material was injected with triphasic or quadriphasic pattern for obtaining homogeneously filled chamber images with decreased artifacts. We used dual head power injector at a dose of 1.5–2mL/ kg and a flow rate of 0.3–3.0 mL/s. Scan delay was determined using a bolus tracking system with ROI within the LV.

## Study design

Existing cardiac CT scans were reconstructed using adaptive statistical iterative reconstruction-V (ASiR-v) and performed using a level of 80% blending with edge contrast reconstruction according to in-hospital agreement, using a 512-slice CT scan (Revolution; GE Healthcare, Milwaukee, USA). After the DLIR of this vendor was approved by the Food and

Drug Administration as a commercialized deep learning reconstruction method, adaptation of DLIR (TrueFidelity; GE Healthcare, Milwaukee, USA) was performed at our center in December 2019. Subsequently, all pediatric cardiac protocol CT scans were reconstructed using ASiR-V and all three levels of DLIR: low, medium, and high.

For an exact comparison, we included pediatric patients with underlying congenital heart disease who underwent pediatric cardiac CT with same vender after adaptation of DLIR. We included cases which canbe reconstructed by using both ASiR-V and DLIR. We excluded studies with severe artifacts including motion or metallic artifacts that can make measurements difficult or insufficient for reconstruction.

## Clinical data and radiation dose

Clinical data, including age at the examination date, sex, body weight, height, and body mass index, were recorded for each patient. Patients' underlying cardiac diseases and recent surgical records were also collected. The duration between the most recent operation and the CT examination was calculated. The reasons for the examinations were also recorded.

We collected the estimated CT dose index volume and dose-length product data for all scans. These data were automatically calculated using a CT machine during the examination and displayed on a picture archiving and communication systems (PACS).

## Image analysis

All CT image datasets were displayed on the PACS workstation (G3 PACS; Infinitt Inc.) with a mediastinal setting of 800 Hounsfield unit (HU) window width and 150 HU window level.

## Quantitative image quality

Objective analysis of quantitative image quality was performed by calculating the signal-to-noise ratio (SNR) and contrast-to-noise ratios (CNR). The average signal attenuation was measured using the mean HU within the region of interest (ROI). The standard deviation within the ROI was calculated and considered a noise parameter.

To reduce bias, two pediatric radiologists (H.-H.C. and S.K.Y, each with 10 years of experience with CT) were blinded to all clinical data, medical information, and radiation dose data.

Each observer drew regions of interest in each area and level for each image set, and the mean value was used for the final analysis. We measured the noise as the average standard deviation calculated using the mean value of each drawn ROI. At the T7–8 level, four ROIs were measured in the right ventricle (RV), left ventricle (LV) chamber, interventricular septum, and paravertebral muscles. At the T2–4 level, two ROIs were measured in the ascending and descending aortas.

For single-ventricle patients, the ROI for the chamber was measured at the right and left sides of a single ventricle. The ROI was selected for muscle in apical area of the cardiac muscle or small portion of residual interventricular septum.

To reduce bias and variability, averaging data of the standard deviation of each area were calculated after drawing three different 10-mm$^2$ circular ROIs in the area where we wanted to measure.

Using the measured data, the SNR of the RV, LV chamber, interventricular septum, paravertebral muscle, and ascending and descending aortas were calculated using the following equation:

$$SNR = \frac{Mean\ HU\ of\ tissue\ in\ ROI}{SD\ of\ HU\ in\ ROI}$$

**Table 1. Visual grade analysis for qualitative analysis.** A) Set of structures established as diagnostic requirements. B) Scores for visibility of the structures in relation to reference structure.

| Organ | Structure |
|---|---|
| **Left/ Right coronary artery** | Ostium<br>At 1.5 cm<br>Distality |
| **Cardiac cavity** | Septum<br>LV/RV lateral wall |
| **Aorta** | Aortic root<br>Aortic cross<br>Ascending/ Descending aorta |
| **Pulmonary artery** | Pulmonary trunk<br>Left/Right pulmonary artery<br>Left/Right pulmonary distality |
| **Lung** | Interstitium |
| **Relative score** | **Visibility** |
| +3 | Definitely better |
| +2 | Better |
| +1 | Slightly better |
| 0 | Equal |
| -1 | Slightly lower |
| -2 | Lower |
| -3 | Definitely lower |

where Mean and SD refer to the mean density and standard deviation of the Hounsfield numbers, respectively, and ROI refers to region of interest of measured level.

The CNR was calculated in four areas: RV, LV chamber, and ascending and descending aortas.

CNR was calculated for each of the four areas using the following equation:

$$CNR = \frac{Mean\ HU\ chamber - Mean\ HU\ muscle}{SD\ of\ HU\ in\ chamber}$$

where Mean and SD refer to the mean density and standard deviation of the Hounsfield numbers, respectively. The density of the interventricular septum was used as the muscle density for the calculation.

**Table 2. Comparison of SNR of image sets (mean±standard deviation).**

| SNR | ASiR-V | DLIR | | | P-value* | | |
|---|---|---|---|---|---|---|---|
| | | High | Med | Low | ASiR-V vs H | ASiR-V vs M | ASiR-V vs L |
| **RV** | 15.98±6.85 | 18.83±8.46 | 13.84±6.04 | 12.58±5.43 | 0.103 | 0.025 | 0.046 |
| **LV** | 18.45±7.20 | 19.01±8.17 | 14.36±4.89 | 12.73±4.88 | 0.717 | <0.001 | <0.001 |
| **Ascending aorta** | 21.61±6.84 | 22.61±8.37 | 18.00±6.43 | 16.39±7.21 | 0.147 | <0.001 | <0.001 |
| **Descending aorta** | 18.70±6.17 | 20.03±7.01 | 16.44±5.50 | 14.97±5.78 | 0.011 | <0.001 | <0.001 |
| **Interventricular septum** | 5.46±2.88 | 3.91±2.12 | 3.77±2.06 | 3.17±1.46 | <0.001 | <0.001 | <0.001 |
| **Paravertebral septum** | 3.67±1.35 | 3.37±1.12 | 2.66±0.96 | 2.35±0.89 | 0.046 | <0.001 | <0.001 |

DLIR Deep Learning Image Reconstruction adapted image sets with strength level (high, medium, low), Asir-V adaptive statistical iterative reconstruction, SNR signal-to-noise ratio, RV right ventricle, LV left ventricle

* post hoc test P<0.05 between Asir-V and each strength of DLIR

**Table 3. Comparison of CNR of image sets (mean±standard deviation).**

| CNR | ASiR-V | DLIR | | | P-value* | | |
|---|---|---|---|---|---|---|---|
| | | High | Med | Low | ASiR-V vs H | ASiR-V vs M | ASiR-V vs L |
| RV | 18.02±6.47 | 18.83±8.46 | 14.84±6.04 | 13.58±5.43 | 0.205 | <0.001 | <0.001 |
| LV | 18.45±7.20 | 19.01±8.17 | 14.36±4.89 | 12.73±4.88 | 0.717 | <0.001 | <0.001 |
| Ascending aorta | 19.40±6.31 | 20.09±7.61 | 15.96±5.83 | 14.57±6.59 | 0.365 | <0.001 | <0.001 |
| Descending aorta | 16.68±5.63 | 17.74±6.42 | 14.53±4.99 | 13.26±5.31 | 0.030 | <0.001 | <0.001 |

DLIR Deep Learning Image Reconstruction adapted image sets with strength level (high, medium, low), Asir-V adaptive statistical iterative reconstruction, CNR contrast-to-noise ratio, RV right ventricle, LV left ventricle

*post hoc test P<0.05 between Asir-V and each strength of DLIR

## Analysis for image sharpness

Edge rise distance (ERD) has been used in previous studies to evaluate CT image quality [21–24]. Axial CT images were obtained at T8. For measurement of ERD, we used a commercialized program to measure the line density profile (Image J program) with a 1 cm reference line drawn from the center of the descending aorta to the adjacent lung parenchyma, according to a previous report [17, 21].

Subsequently, we drew a graph based on the measured data and calculated the ERD and angle between 25% and 75% of the line density profile using the statistical program R. The ERD and angle values were compared among the reconstruction algorithms. All measurements were performed by one of the authors (H-H.C. with 11 years of experience in pediatric radiology), and the measurements were repeated three times for each image to reduce measurement variability.

## Qualitative image quality

Qualitative image quality was rated with reference to the visualization of structures for the diagnostic quality of pediatric cardiac CT [15] with visual grading analysis (VGA) scores to grade visibility (Table 1). Qualitative image quality analysis was performed by two attending pediatric radiologists (H.-H.C and S.M.L., each with 11 years of clinical experience). To reduce bias, all identifying data were removed from each image set, and the image sets were reordered randomly before analysis. Interobserver agreement between the two radiologists was also calculated.

## Statistical analysis

To compare the quantitative parameters of the DLIR at high, medium, and low levels with those of the ASiR-V group, a paired Student's t-test was used. To compare the results of the

**Table 4. Comparison of sharpness of image sets.**

| | ASiR-V | DLIR | | | P value* | | |
|---|---|---|---|---|---|---|---|
| | | High | Med | Low | ASiR-V vs H | ASiR-V vs M | ASiR-V vs L |
| Distance 25–75% (mm) | 47.00±17.45 | 44.92±13.64 | 48.90±16.07 | 51.58±23.07 | 0.048 | 0.182 | 0.144 |
| Angle 25–75% | 75.19±3.20 | 77.06±3.28 | 75.66±3.06 | 74.39±2.75 | 0.002 | 0.061 | <0.001 |

DLIR Deep Learning Image Reconstruction adapted image sets with strength level (high, medium, low), Asir-V adaptive statistical iterative reconstruction

*post hoc test P<0.05 between Asir-V and each strength of DLIR

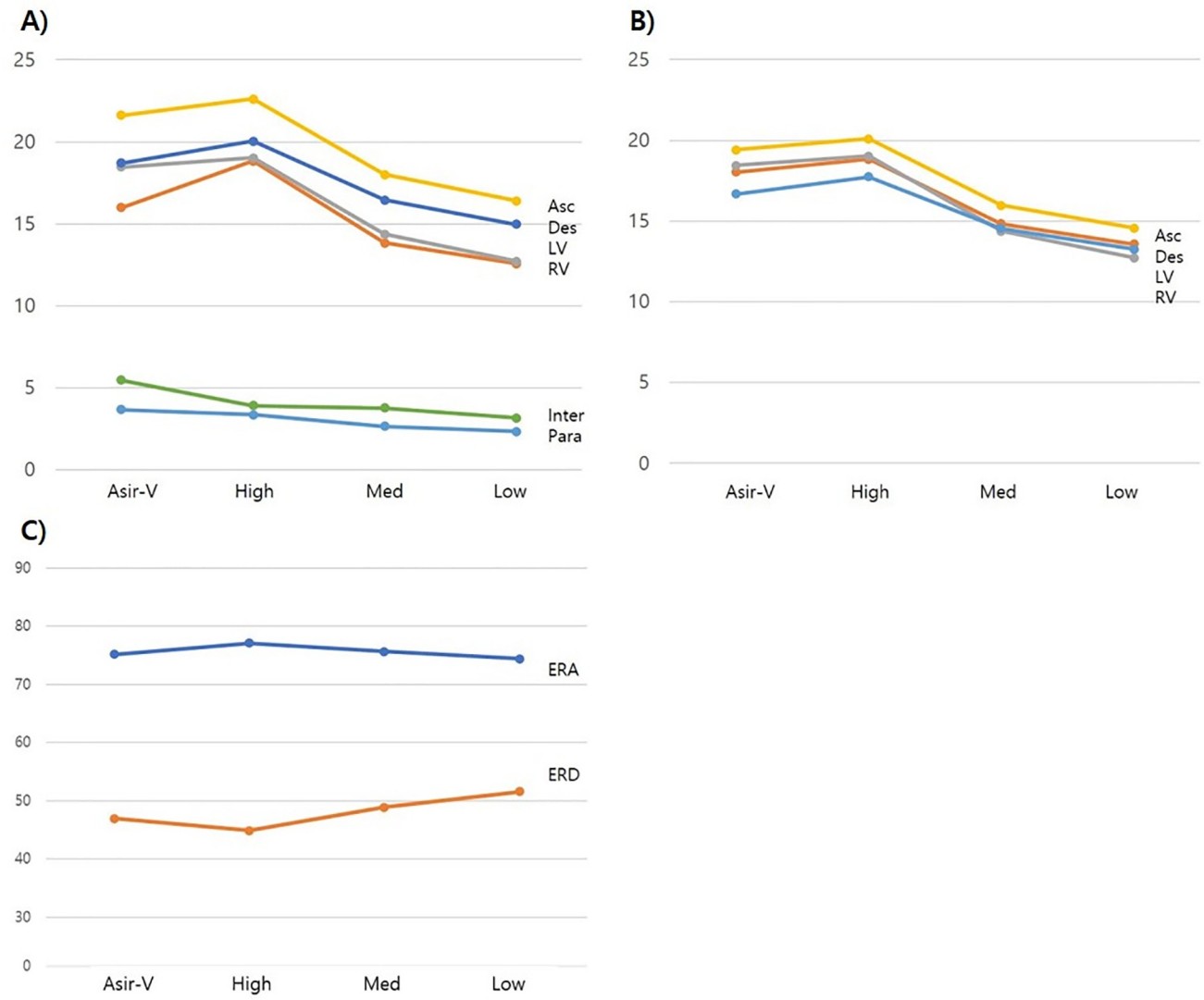

**Fig 1.** Graphs for SNR (A), CNR (B), ERD (C) and ERA (D) among reconstruction methods. Asir-V adaptive statistical iterative reconstruction, DLIR Deep Learning Image Reconstruction adapted image sets with strength level (high, medium, low), Asc Ascending aorta, Des Descending aorta, LV left ventricle, RV right ventricle, Inter Interventricular septum, Para Paravertebral septum.

qualitative analysis of image quality, the results of DLIR high, medium, and low levels were compared with those of the ASiR-V group using the Kruskal-Wallis test and analysis of variation (ANOVA). Inter-observer agreement was assessed using weighted Cohen's kappa statistics. All statistical analyses were conducted using commercially available software (SPSS Statistics, Version 19.0; IBMm, Armonk, NY, USA).

## Results

### Clinical data and radiation dose

Between January 2020 and December 2022, 123 pediatric cardiac CT scans were included in this study. Among these, a total of 109 patients (48 females, 61 males; mean age, 102.1 months) underwent pediatric protocol cardiac CT. Underlying cardiac diseases included pulmonary

**Table 5. Comparison of qualitative analysis of image sets (mean scores).**

|  | DLIR | | | P value* |
|---|---|---|---|---|
|  | High | Med | Low |  |
| RCA OS | 1.70 | 0.19 | -0.98 | <0.001 |
| 1.5 | 1.54 | 0.15 | -1.16 | <0.001 |
| Dis | 0.68 | 0.22 | -1.06 | <0.001 |
| LCA OS | 1.62 | 0.18 | -1.07 | <0.001 |
| 1.5 | 1.54 | 0.13 | -1.05 | <0.001 |
| Dis | 1.47 | 0.24 | -1.07 | <0.001 |
| Cavity Septum | 1.58 | 0.24 | -0.94 | <0.001 |
| LV | 1.63 | 0.14 | -1.13 | <0.001 |
| RV | 1.39 | 0.15 | -0.96 | <0.001 |
| Aorta Root | 1.58 | 0.18 | -1.05 | <0.001 |
| Cross | 1.76 | 0.12 | -1.01 | <0.001 |
| Ascending | 1.46 | 0.21 | -0.99 | <0.001 |
| Descending | 1.65 | 0.17 | -1.14 | <0.001 |
| PA Trunk | 1.66 | 0.15 | -1.04 | <0.001 |
| LPA | 1.58 | 0.14 | -1.06 | <0.001 |
| RPA | 1.62 | 0.22 | -1.15 | <0.001 |
| LP dis | 1.68 | 0.15 | -0.97 | <0.001 |
| RP dis | 1.59 | 0.27 | -1.15 | <0.001 |
| Lung | -1.77 | 0.13 | -1.11 | <0.001 |

DLIR Deep Learning Image Reconstruction adapted image sets with strength level (high, medium, low), Asir-V adaptive statistical iterative reconstruction, RCA right coronary artery, OS ostium, Dis distality, LCA left coronary artery, LV left ventricle, RV right ventricle, PA pulmonary, LPA left pulmonary artery, RPA right pulmonary artery, LPD left pulmonary artery distality, RPD right pulmonary artery distality

*post hoc test P<0.05 between Asir-V and each strength of DLIR

stenosis (n = 15), pulmonary atresia with VSD (n = 5), Tetrology of Fallot (n = 23), functional single ventricle (n = 30), interrupted aortic arch (n = 9), coarctation of the aorta (n = 3), and other diseases (n = 38).

CT scans were performed before the operation in 63 patients, during routine follow-up in 40 patients, and during evaluation of postoperative complications in six patients.

## Quantitative analysis

The results of the quantitative image analysis are presented in Tables 2 and 3. A gradual improvement in the SNR was noted among the strength levels of the DLIR in sequence from low to high. When compared with ASiR-V in all measured areas except muscles, high-level DLIR showed better SNR without significant difference. The SNR of low, medium-level DLIR showed a significantly lower $P$ value than that of ASiR-V in all measured areas except RV (P <0.05).

The CNR of the RV, LV, and ascending and descending aortas showed an improvement in strength levels of the DLIR in sequence from low to high. Compared with ASiR-V, high-level DLIR showed better CNR without significant difference. The CNR of low, medium-level DLIR showed a significantly lower $P$ value than that of ASiR-V ($P<0.05$) at all levels.

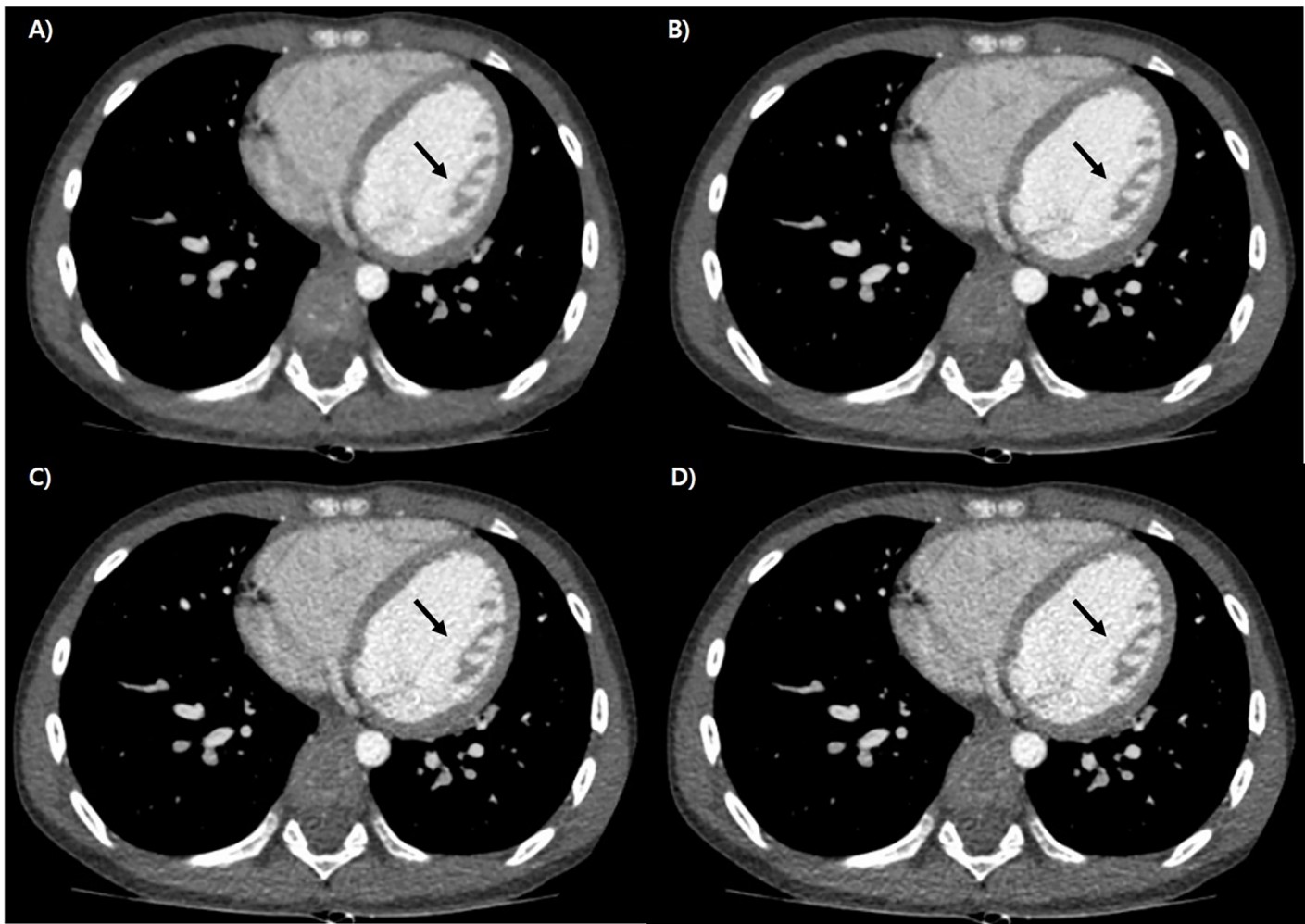

**Fig 2.** Representative case of cardiac CT of 7 years old boy, which was reconstructed by ASiR -V (A), DLIR- High (B), Med (C) and Low (D). There noted increment of differentiation of intraventricular muscles (black arrow) with decreased noise in contrast filled cavity.

### ERD

The distance between the 25% and 75% levels decreased as the DLIR level increased (Table 4). The level of high-level DLIR was significantly higher than that of ASiR-V ($P$ = 0.048). The angle between 25% and 75% increased as the DLIR level increased. The value of high-level DLIR was significantly higher than that of ASiR-V ($P$ = 0.002). The results of quantitative analysis are also shown in graphs (Fig 1).

### Qualitative analysis of image quality

The results of the qualitative image analysis are presented in Table 5. Qualitative analysis of image quality scored using VGA revealed sequentially increased scores among the strength levels of DLIR from low to high, indicating improvement in the detection of anatomical structures (Figs 2–4). All scores were significantly better in the higher-level DLIR ($P$ = 0.001).

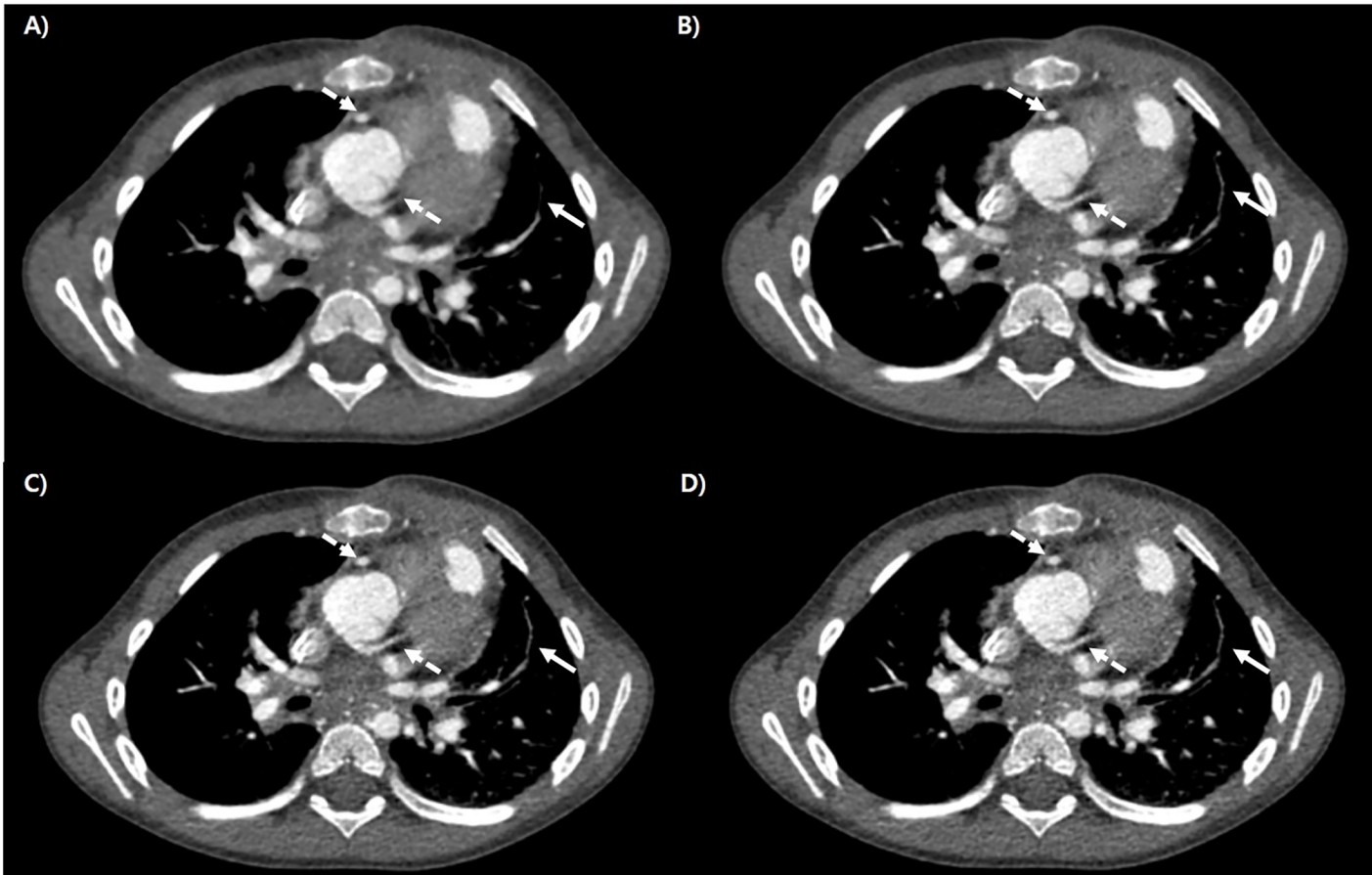

**Fig 3.** Representative case of cardiac CT of 4 months old boy, which was reconstructed by ASiR -V (A), DLIR- High (B), Med (C) and Low (D). There noted increment of differentiation of coronary arteries (dot white arrows). Distal branch of left pulmonary artery is more well visualized in high level DLIR (white arrows).

## Discussion

Recent studies on the image quality of cardiac CT scans have focused on the adaptation of post-processing reconstruction methods, including many iterative reconstructions such as SAFIRE and ASiR [12–14, 25] in both pediatric and adult patients [15, 26, 27]. Because the DLR algorithm is a recently developed reconstruction method that lacks many clinical results [18, 19, 28], the results of this study might be helpful in substituting DLIR, especially for pediatric cardiac CT. Compared to recent studies using the DLIR in pediatric CT [9, 29], this study has the advantage of conducting a more objective assessment with more objective parameters in a larger population.

The results of this study indicate that high-level DLIR can achieve ~~significantly higher~~ as highe SNR and CNR ~~like~~ as those of ASiR-V. As ASiR-V is an established reconstruction method for pediatric CT studies according to previous studies [12, 13, 15, 25], high-level DLIR can be used as an alternative method for pediatric cardiac CT. Although CNR and SNR showed high values in this study, but they didn't show significantly increased value compared to ASiR-V, as noted in previous study [9] which used different blending factor for ASiR-V. So, there needed further studies using different blending factors for ASiR-V. H

However, the measured ERD was significantly shorter with a larger ERA in high-level DLIR than in ASiR-V, indicating that a sharper graph was drawn with a high-level DLIR. The

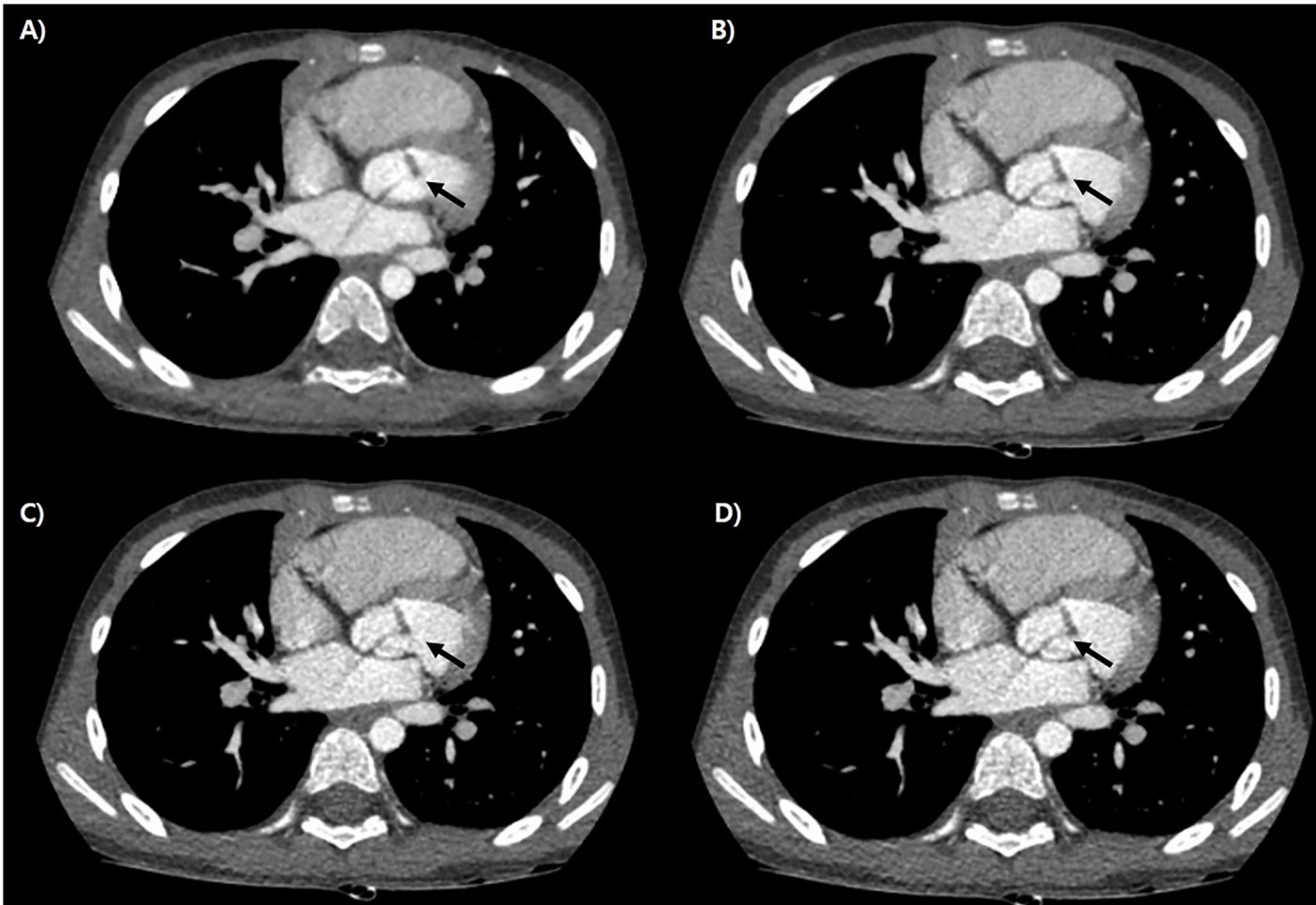

**Fig 4.** Representative case of cardiac CT of 7 months old boy, which was reconstructed by ASiR -V (A), DLIR- High (B), Med (C) and Low (D). There noted increment of differentiation of aortic valve leaflets (black arrows).

measurement of ERD and ERA in this study is significant because these methods can be more objective in proving that the sharpness of the DLIR CT scan is better than that of ASiR-V using methods that have not been used as comparison methods in a previous study [9]. ERD and ERA can be used as the most accurate indicators of image sharpness by measuring distances of 25–75% and angles of 25–75%, which would be shorter and larger on the sharper border of the changing density of each reconstructed image [17, 21–24]. In particular, considering that the part to be checked in pediatric cardiac CT is the structure formed by the vessels, atria, and ventricles, which are filled with contrast media, this difference in sharpness plays an important role in evaluation of anatomical structures in pediatric cardiac CT [21].

Qualitative analysis measuring the VGA also showed significantly better scores for the higher-level DLIR. VGA scoring is important because a better score indicates better delineation of the distal small part of the thoracic structures, which is important for evaluating small anatomical structures in pediatric patients [15]. Especially in patients with complex congenital heart disease, clear visualization of small vessels and muscle fibers is very important for the pre- or post-operative evaluation of pediatric cardiac CT scans.

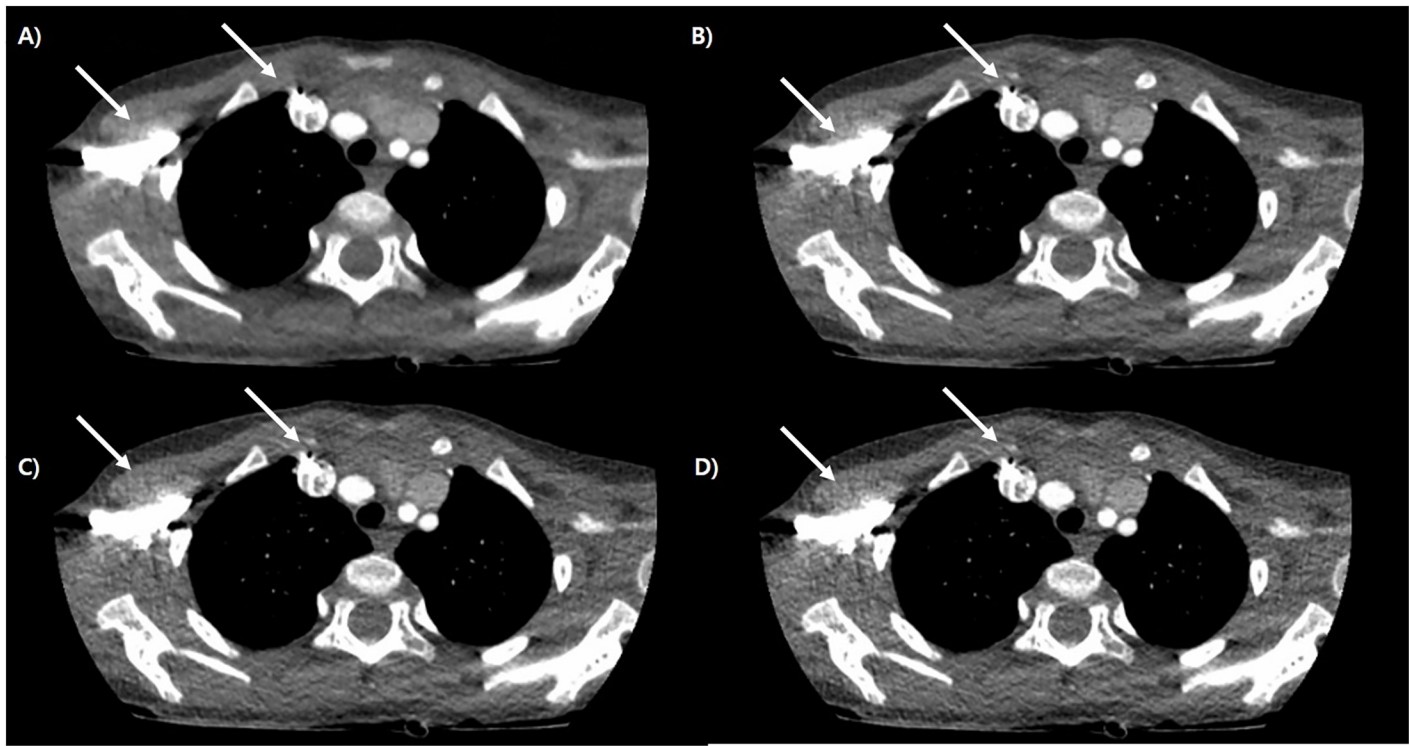

**Fig 5.** Representative case of cardiac CT of 4 years old boy, which was reconstructed by ASiR -V (A), DLIR- High (B), Med (C) and Low (D). There noted distortion artifacts in all 4 reconstruction methods and artifact due to contrast showed more severe on DLIR methods than ASiR-V (white arrows).

We also evaluated the distortion artifact, which is a characteristic artifact of DLIR, as suggested in previous studies [16, 17]. This distortion artifact, noted as a 'checkered pattern' artifact [16, 17], is noted in our cases, but it was also noted in the ASiR-V (Fig 5); therefore, this artifact is not a typical reconstruction for the DLIR. This finding supports the hypothesis of a previous report that the artifact may be due to the CT scanner hardware or reconstruction algorithm [16, 17].

Our previous study about pediatric brain CT scan revealed that there noted no improvement of artifact or even worsened after adaptation of DLIR in some age groups [30]. In this study some cases also showed worsening of artifacts by contrast in high level DLIR than in ASiR-V (Fig 5), unless, which do not effect on the evaluating anatomy. And it can be thought to be by shortness of artifacts for training this reconstruction method [30].

This study had several limitations. The sample included patients of various age groups. Hence, we evaluated the image quality of the two reconstruction methods. The results showed similar trends among different age groups. Due to the retrospective nature of this study, we could not compare FBP and DLIR. In terms of lesion detection, lesions on pediatric cardiac CT are not significantly affected by reconstruction methods; therefore, the comparison of lesion detection was not meaningful in these patients.

## Conclusion

According to this study, high-level DLIR showed better qualitative and quantitative image quality than ASiR-V and other levels of DLIR. Although ASiR-V is an established reconstruction method for pediatric CT studies, high-level DLIR can be used as an alternative to pediatric cardiac CT.

## Supporting information

**S1 File. Included data file.**
(ZIP)

## Author Contributions

**Conceptualization:** Hyun-Hae Cho, Sun Kyoung You.

**Data curation:** Hyun-Hae Cho.

**Formal analysis:** So Mi Lee.

**Investigation:** Hyun-Hae Cho, So Mi Lee.

**Methodology:** Hyun-Hae Cho, So Mi Lee, Sun Kyoung You.

**Software:** Hyun-Hae Cho, Sun Kyoung You.

**Supervision:** Sun Kyoung You.

**Validation:** So Mi Lee.

**Visualization:** So Mi Lee.

**Writing – original draft:** Hyun-Hae Cho.

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
