## [Decision Letter · Decision Letter 0]

4 Apr 2024

PONE-D-24-04756Assessment of deep learning image reconstruction (DLIR) on image quality in pediatric cardiac CT datasetsPLOS ONE

Dear Dr. Cho,

Thank you for submitting your manuscript to PLOS ONE. After careful consideration, we feel that it has merit but does not fully meet PLOS ONE’s publication criteria as it currently stands. Therefore, we invite you to submit a revised version of the manuscript that addresses the points raised during the review process.

We look forward to receiving your revised manuscript.

Kind regards,

Ashraful Hoque

Academic Editor

PLOS ONE

“This work was supported by the Ewha Womans University Research Grant of 2021.”

3. We are unable to open your Supporting Information file [Data.zip]. Please kindly revise as necessary and re-upload.

4. Please ensure that you include a title page within your main document. You should list all authors and all affiliations as per our author instructions and clearly indicate the corresponding author.

6. We note that your Data Availability Statement is currently as follows: [All relevant data are within the manuscript and its Supporting Information files.]

Reviewers' comments:

Reviewer's Responses to Questions

**Comments to the Author**

1. Is the manuscript technically sound, and do the data support the conclusions?

Reviewer #1: Yes

Reviewer #2: Yes

2. Has the statistical analysis been performed appropriately and rigorously? 

Reviewer #1: Yes

Reviewer #2: Yes

3. Have the authors made all data underlying the findings in their manuscript fully available?

Reviewer #1: No

Reviewer #2: Yes

4. Is the manuscript presented in an intelligible fashion and written in standard English?

Reviewer #1: No

Reviewer #2: Yes

5. Review Comments to the Author

Reviewer #1: - The overall length of the background is adequate; however, this section needs significant improvement. P3L47-53 is redundant, and it's generally not stylistically pleasing to have only two references, one of which is repeated in two different sentences.

- What is the role of DLIR? Why is it better than iterative reconstructions? The latter have been used in both CCTA and calcium scoring, but they have intrinsic flaws and have not been mentioned at all.

- 10.1097/RTI.0000000000000340

- 10.1016/j.ejrad.2017.03.011

- 10.1007/s00330-019-06359-6

- 10.2214/AJR.10.4285

- Recent publications on this specific topic (DLIR and CCTA) have been overlooked. They could increase the value of the introduction or discussion. Please consider adding them.

- 0.1007/s11547-023-01607-8

- 10.1016/j.jcct.2020.01.002

- 10.1007/s00330-021-08367-x

- 10.1007/s00330-021-08424-5

• A paragraph on image acquisition must be written with all the information to be reproducible. Information on the injection of contrast media must also be provided.

• Inclusion and exclusion criteria must be clearly stated.

• What was used as the muscle parameter in patients with single ventricle? In these patients, the current CNR formula may not have been applicable.

• This ERD parameter is kind of obscure to me, I suggest performing a "traditional" subjective image quality analysis like almost every radiology manuscript does, readers are surely more familiar with it.

• Results: please double-check the number of patients: the sum of different pathologies is greater than the total, and the number of patients with "other diseases" is also missing.

• Results: simply indicate the actual p-values, not < 0.05: it is implicit that a statistically significant value is less than 0.05.

• "Compared to recent studies using DLIR, this study has the advantage of conducting a more objective assessment of pediatric cardiac CT scans in a larger population." Authors cannot write such a statement without providing any citation. What are these studies? What is their population? Why would this study be more objective? I suggest toning down statements like these.

• In the discussion, there are many general statements without citations, and the results are not properly discussed, as they should be.

Reviewer #2: Although I have fully confirmed the authors' efforts for the results of this study, there are several maor comments.

1. ASiR-V and DLIR are thought to be different kinds of image reconstruction methods. However, the authors did not make a distinction between these two reconstruction methods. For example, in Table 2, the authors would need to separate DLIR and ASiR-V.

2. I would like to emphasize in the purpose of the paper what differentiates it from existing studies. In fact, there is a paper that is very similar to this paper. ( Yoon et al. Image quality assessment of pediatric chest and abdomen CT by deep learning reconstruction, BMC Medical Imaging, 2021)

6. PLOS authors have the option to publish the peer review history of their article (what does this mean?). If published, this will include your full peer review and any attached files.

Reviewer #1: No

Reviewer #2: No

---

## [Author Response · Author response to Decision Letter 0]

25 May 2024

PONE-D-24-04756

Assessment of deep learning image reconstruction (DLIR) on image quality in pediatric cardiac CT datasets

1. Reviewer #1. 

#1 Comment

 - The overall length of the background is adequate; however, this section needs significant improvement. P3L47-53 is redundant, and it's generally not stylistically pleasing to have only two references, one of which is repeated in two different sentences.

Author response : Thank you for your comments. We revised background part according to the comments and added suitable citations. 

#2 Comment 

- What is the role of DLIR? Why is it better than iterative reconstructions? The latter have been used in both CCTA and calcium scoring, but they have intrinsic flaws and have not been mentioned at all.

- 10.1097/RTI.0000000000000340

- 10.1016/j.ejrad.2017.03.011

- 10.1007/s00330-019-06359-6

- 10.2214/AJR.10.4285

Author response : Thank you for your comments. Actually DLIR is most recently released post processing tool by GE. And there need some verifications for adaptation for clinical studies. Besides the other studies results, we need to evaluate the adaptation of DLIR for pediatric CHD patients for clinically appreciable or not. 

#3 Comment 

- They could increase the value of the introduction or discussion. Please consider adding them.

- 0.1007/s11547-023-01607-8

- 10.1016/j.jcct.2020.01.002

- 10.1007/s00330-021-08367-x

- 10.1007/s00330-021-08424-5

Author response : Thank you for your comments. So, we added them for introduction and discussion. 

#4 Comment 

• A paragraph on image acquisition must be written with all the information to be reproducible. Information on the injection of contrast media must also be provided.

Author response : Thank you for your comments. We added the imaging part including the pediatric cardiac CT protocol of our center. 

#5 Comment 

• Inclusion and exclusion criteria must be clearly stated.

Author response : Thank you for your comments. We revised and added it. 

#6 Comment 

• What was used as the muscle parameter in patients with single ventricle? In these patients, the current CNR formula may not have been applicable.

Author response : Thank you for your comments. We revised and added it.

#7 Comment 

• This ERD parameter is kind of obscure to me, I suggest performing a "traditional" subjective image quality analysis like almost every radiology manuscript does, readers are surely more familiar with it.

Author response : Thank you for your comments. Actually, ERD and ERA can be unfamiliar, but it is recently widely used tool for calculation of sharpness. We added some citations of previous studies using ERD and ERA for understanding. 

#8 Comment 

Results: please double-check the number of patients: the sum of different pathologies is greater than the total, and the number of patients with "other diseases" is also missing.

Author response : Thank you for your comments. We made some mistakes for record the number of each diseases. And we correct them. 

#9 Comment 

• Results: simply indicate the actual p-values, not < 0.05: it is implicit that a statistically significant value is less than 0.05.

Author response : Thank you for your comments. We added actual P values for ERD. And those of SNR and CNR is hard to added because P values noted with variable values in each areas. So, we added those values in table 2 And 3. 

#10 Comment

• "Compared to recent studies using DLIR, this study has the advantage of conducting a more objective assessment of pediatric cardiac CT scans in a larger population." Authors cannot write such a statement without providing any citation. What are these studies? What is their population? Why would this study be more objective? I suggest toning down statements like these.

Author response : Thank you for your comments. Recent study was citated. And it was focused on the CNR and SNR for image analysis, but this study added VGA, ERD and angle for more exact evaluation. 

#11 Comment 

• In the discussion, there are many general statements without citations, and the results are not properly discussed, as they should be.

Author response : Thank you for your comments. We revised the discussion part following your comments. 

2. Reviewer #2.

Although I have fully confirmed the authors' efforts for the results of this study, there are several maor comments.

#1 Comment 

- ASiR-V and DLIR are thought to be different kinds of image reconstruction methods. However, the authors did not make a distinction between these two reconstruction methods. For example, in Table 2, the authors would need to separate DLIR and ASiR-V.

Author response : Thank you for your comments. We revised the tables according to your comments. 

#2 Comment

 I would like to emphasize in the purpose of the paper what differentiates it from existing studies. In fact, there is a paper that is very similar to this paper. (Yoon et al. Image quality assessment of pediatric chest and abdomen CT by deep learning reconstruction, BMC Medical Imaging, 2021)

Author response : Thank you for your comments. Actually, the difference occurs in the objective group between the studies. The pediatric cardiac CT scan is focused on the separating of each structures, so the discrimination and contrast is more important in them. 

We revised the purpose and discussion part according to the comments.

---

## [Editor Report · Acceptance letter]

21 Jun 2024

PONE-D-24-04756R1 

PLOS ONE

Dear Dr. Cho, 

I'm pleased to inform you that your manuscript has been deemed suitable for publication in PLOS ONE. Congratulations! Your manuscript is now being handed over to our production team.

Kind regards, 

on behalf of

Dr. Ashraful Hoque 

Academic Editor

PLOS ONE